# Status of countrywide laboratory services quality and capacity in primary healthcare facilities in Tanzania: Findings from Star Rating Assessment

**Erick Kinyenje** [1,☯]*, **Ruth R. Ngowi** [1,☯], **Yohanes S. Msigwa** [1], **Joseph C. Hokororo** [1], **Talhiya A. Yahya** [2], **Chrisogone J. German** [1], **Akili Mawazo** [3], **Mohamed A. Mohamed** [4,5], **Omary A. Nassoro** [1], **Mbwana M. Degeh** [1], **Radenta P. Bahegwa** [1], **Laura E. Marandu** [1], **Syabo M. Mwaisengela** [1], **Lutengano W. Mwanginde** [6], **Robert Makala** [7], **Eliudi S. Eliakimu** [1]

**1** Health Quality Assurance Unit, Ministry of Health, Dodoma, Tanzania, **2** Management Science for Health, Dar es Salaam, Tanzania, **3** Department of Microbiology and Immunology, School of Medicine, Muhimbili University of Health and Allied Sciences, Dar es Salaam, Tanzania, **4** Tanzania Field Epidemiology and Laboratory Training Programme (TFELTP), Dar es Salaam, Tanzania, **5** East Central and Southern Africa Health Community, Arusha, Tanzania, **6** Tanzania Police Force, Forensic Bureau, Dar es Salaam, Tanzania, **7** Regional Administrative Secretary's Office—Regional Health Management Team, Manyara, Tanzania

☯ These authors contributed equally to this work.

\* kinyenje2003@yahoo.com

**Data Availability Statement:** Data for this study can be requested from Principal Secretary of Ministry of Health through ps@afya.go.tz

## Abstract

Accurate disease diagnosis relies on a well-organized and reliable laboratory system. This study assesses the quality of laboratory services in Tanzania based on the nationwide Star Rating Assessment (SRA) of Primary Healthcare (PHC) facilities conducted in 2017/18. This cross-sectional study utilized secondary data from all the country's PHC facilities stored in the SRA database. Laboratory service quality was assessed by aggregating scores as percentages of the maximum achievable score across various indicators: dedicated laboratory department/room, adequate equipment, staffing levels, adherence to testing protocols, establishment of turnaround times, internal and external quality controls, and safety and supplies management. Scores equal to or exceeding 80% were deemed compliant. Multiple linear regression was used to determine the influence of facility characteristics (level, ownership, location, staffing) on quality scores, with statistical significance set at p < 0.05. The study included 6,663 PHC facilities (85.9% dispensaries, 11% health centers, 3.2% hospital-level-1), with the majority being public (82.3% vs. 17.7%) and located in rural areas (77.1% vs. 22.9%). On average, facilities scored 30.8% (SD = 35.7), and only 26.6% met staffing requirements. Compliance with quality standards was higher in private (63% vs. 19%, p<0.001) and urban facilities (62% vs. 16%, p<0.001). More than half of the facilities did not meet either of the eight quality indicators. Quality was positively linked to staffing compliance (Beta = 5.770) but negatively impacted by dispensaries (Beta = -6.342), rural locations (Beta = -0.945), and public ownership (Beta = -1.459). A score of 30% falls significantly short of the national target of 80%. Improving laboratory staffing levels at PHC facilities could improve the quality of laboratory services, especially in public facilities that are

**Funding:** The authors received no specific funding for this work.

**Competing interests:** The authors have declared that no competing interests exist.

based in rural areas. There is a need to further strengthen laboratory services in PHC facilities to ensure the quality of laboratory services and clients' satisfaction.

## Introduction

In the past two decades (2000–2020), Tanzania has made significant efforts to strengthen its health system as part of the then health sector reforms that started in 1994; coupled with interventions for health system strengthening following the World Health Organization (WHO) guidance [1, 2]. The efforts included the construction and rehabilitation of health infrastructure at all levels of health services delivery, procurement of equipment, and capacity building of health personnel. The efforts have enabled improvement in diagnostic availability in the primary health care (PHC) facilities. For example, a recent analysis has shown that diagnostic availability in dispensaries and health centres in Tanzania, have increased by 6 percentage points and 7.8 percentage points respectively from 2006 to 2014 although there are variations across the country [3].

Specific efforts that aimed at strengthening and improving the quality of laboratory services provided included the following: (i) harmonization and standardization of laboratory equipment in various levels of health services delivery including PHC facilities (mainly district hospitals and health centers) in order to address the challenges on the areas of procurement of reagents, maintenance of equipment, and quality assurance; [4, 5] (ii) establishment of the National Health Laboratory Quality Assurance and Training Centre since 2008 with "*the purpose of improving quality of laboratory services all over the country*" currently referred to National Public Health Laboratory; [6] (iii) development and implementation of the "*National Health Laboratory Strategic Plan 2009–2015*"; [7] (iv) implementation of the World Health Organization African Region Stepwise Laboratory Improvement Process Towards Accreditation (WHO AFRO SLIPTA) in 2009 and implementation of the laboratory training program (Strengthening Laboratory Management Towards Accreditation (SLMTA)), in 2010; [8] and (v) development of the Point of Care Testing (POCT) Certification Framework in October 2017 aiming at improving the quality of HIV testing services in health facilities [9]. In Africa, Laboratory services especially in the area of HIV have improved significantly with potential for supporting country systems in future global health threats [10] since the Maputo Declaration of 2008 [11].

Importance of a strong PHC system has been further demonstrated by the ongoing coronavirus disease of 2019 (COVID-19) pandemic in which its effects have shown clearly the need for a strong PHC as a means to uphold equity and strengthen capacity to respond to emergencies [12]. Laboratory services in PHC facilities have been shown to enhance performance of PHC facilities [13]. Also, the framework produced by the Lancet Global Health Commission for High Quality Health Systems in Sustainable Development Goals Era, shows that improving laboratory services in PHC facilities is an essential element for ensuring "*competent care and systems*" which is one of the components of processes of care in the framework [14].

Access to laboratory services at the district level in Tanzania has been shown to be inequitable in the sense that in some areas, people have longer travel time to access a nearby facility for the services [15]. An analysis of data from 10 countries including Tanzania, has shown that only 199 (2%) of the facilities investigated in those countries had all of the diagnostics services [16]. Also, laboratories in PHC facilities have been reported to face a number of challenges in supporting care and treatment services for HIV services which include "*number of qualified*

*personnel*, *staff training on the national guidelines*, *laboratory diagnostic tools and coordination*" [17]. In a study conducted in 2014, basic diagnostic equipment for HIV and diabetes were observed more frequently in hospitals than in health centres and dispensaries [18]. Strengthening laboratory services in PHC facilities will help to capacitate Tanzanian PHC system to tackle the challenge of Non-Communicable Diseases (NCDs) [19]. In Ghana, improvement of POCT diagnostic services in PHC facilities have been recommended as part of improving maternal health services [20].

The Government of Tanzania embarked on an initiative called "*Big Results Now*" in health sector since 2014/2015 with several initiatives among them being performance management of PHC facilities through implementation of Star Rating Assessment (SRA). The SRA was implemented in Tanzania mainland in three period of time: baseline assessment was conducted in the fiscal year 2015/16 all 26 regions; second assessment was conducted in fiscal year 2017/18 in all 26 regions; and third assessment was conducted in fiscal year 2021/22 in 10 regions. The assessments used a set of tools for dispensary, health centre and hospitals at council level, each with 12 service areas that were assessed [21, 22]. Service area number 12 is on clinical support services in which one of the sub—area is laboratory services [23]. Therefore, the purpose of this paper is to describe the status of laboratory services in the PHC facilities during the 2017/18 second assessment in order to recommend measures appropriate for further strengthening of the services towards achievement of the universal health coverage (UHC). The findings of the study will also contribute in understanding the situation of PHC delivery in a way that will be in line with the new health system performance assessment framework to inform policy decisions [24].

The specific objectives of this study are to: determine the average score for all facilities in quality of laboratory services, the proportion of PHC Facilities that comply with staffing level for qualified laboratory personnel; determine the proportion of PHC facilities that comply with other indicators for measuring the quality of laboratory services; and identify the predictors for high scores in indicators that measure the quality of laboratory services among PHC facilities.

## Methods

### Study design

This was a cross-sectional study using secondary data that were collected from all PHC facilities during star rating second assessment in 2017/18. The data are available in the national data platform, i.e., DHIS2.

### Study area

This study presents findings from primary healthcare facilities that are scattered over the Tanzania mainland territory which covers an area of 946,270 square kilometres [25]. With about sixty million population [26] and a GDP per capita (current US$) of 1,099.3 by 2021, the country is regarded as one of the lower-middle-income countries by the World Bank [27]. Half of the population is female and 43% are aged below the age of 15 years [28]. Agriculture is the main economic activity and the largest source of foreign exchange. With an SDG index score of 57.4, the country ranked 130 out of 163 in achieving SDGs by 2022- achieving just two out of 17 goals: climate action and responsible consumption and production [29]. In addition, the country holds the 160th position out of 191 countries in the Human Development Index (HDI) for the year 2021, with a score of 0.549 [30]. The HDI is a measure used by the United Nations to determine the progress of a country towards human development by taking care of

the three dimension indexes: life expectancy index, education index, and Gross national income (GNI) per capita index [31].

## Study population

The healthcare facilities in Tanzania can be grouped into two major categories; PHC facilities and Referral Healthcare Facilities (RHFs). RHFs are referral points for all PHC facilities, these are hospitals that are distributed in all 26 regions of the country. Within each region, there are a number of PHC facilities ranging from dispensaries, health centres to hospitals level 1, in that order of expertise. Dispensaries provide exclusively outpatients' services to about 10,000 populations. Health centres are designated as referral points for dispensaries because they offer a broader range of services including inpatient services and Comprehensive Emergency Obstetric and Newborn Care (CEmONC) to about 50,000 populations. A hospital at the council level (i.e., level 1 hospital) serves about 250,000 population and receives referrals from the low levels [32].

Additional facility characteristics such as the place where the facility is located (urban or rural), and ownership status (private or public) may significantly affect access, management and quality of services delivered. Between 2017 and 2018; SRA was conducted to 7,289 PHC facilities that are unrestrictedly distributed all over the country. PHC facilities constitute about 95% of all healthcare facilities in Tanzania.

**Inclusion criteria.** This study includes all PHC facilities that completed successfully SRA conducted in 2017/18. The SRA aimed to include all PHC facilities; however, those found non-functional were excluded.

**Exclusion criteria.** Any facility whose scores on quality of laboratory services were not found in the SRA database was excluded from the study.

## Data collection and management of SRA database

The SRA database is a part of the DHIS2 platform that is managed by the Health Quality Assurance Unit (HQAU) under Ministry of Health. Paper based tools were used to collect baseline data at 2015/16 assessment while data were collected electronically through DHIS2 during the second and third assessment held in 2017/18 and 2021/22. Since the dataset for baseline were extremely incomplete; we used the second assessment data for this study. The dataset is further divided in 12 quality assessment areas as per SRA tool. Area number 12 is namely clinical support services comprised of two major sections; pharmaceutical services section and laboratory section. The laboratory section contains score data on eight indicators of quality of laboratory services, as presented in Table 1.

## Management of study variables and statistical analysis

Each facility was assigned an overall quality score for laboratory services during the assessment. The score was regarded as the quantitative dependent variable of the study and it was the sum of individual continuous scores from each indicator (Table 1). The facility had a possible maximum score of 12 for dispensaries and 14 for health centres and hospitals (due to additional questions related to blood transfusion facilities and services). For the comparison purpose, the mean scores have been presented in percentages.

The performance in individual indicators for the quality of laboratory services were given as proportions of facilities that scored recommended score of ≥80%. This cut-off point is provided in the National Guidelines for Recognition of Implementation Status of Quality Improvement Initiatives in Health Facilities [38].

**Table 1. Indicators that were used to measure the quality of laboratory services.**

| Indicator | Definition and means of scoring |
|---|---|
| Dedicated department/room for laboratory services | The facility scored "1" for the indicator if had a dedicated department or room for laboratory services, otherwise scored "0". A dedicated room should have been equipped with running water and the basin, plus suitable worktop. |
| Equipped Laboratory | A facility equipped with the necessary equipment scored "1". A list of equipment needed in laboratory can be long [6]; nevertheless, key and few of them were selected by stakeholders and used during SRA. These were: Haemoglobinometer, POC glucometer, tubes for collection of blood, and containers for collection of urine specimens for dispensaries, while microscope and centrifuge were added for health centres and hospitals. |
| Qualified staff for laboratory services | In Tanzania's laboratory profession, there are three distinct cadres: Laboratory Assistants, who undergo two years of training and receive a certificate; Assistant Laboratory Technologists, who engage in three years of training leading to a diploma; and Laboratory Technologists, who earn a degree certificate following a training period of at least three years, plus one year of internship [17]. The indicator scored "1" whenever the facility had met the staffing level for qualified laboratory personnel, otherwise "0". As per Tanzanian *"Staffing Levels for the Ministry of Health and Social Welfare Departments, Health Service Facilities, Health Training Institutions and Agencies of 2014–2019"* [33]; the dispensary was required to have at least one Laboratory Assistant, a health centre to have one Laboratory Technologist and one Assistant Laboratory Technologist. The hospital was to have minimum of five personnel such as two Assistant Laboratory Technologist and three Laboratory Technologists. |
| Essential laboratory tests provided with SOPs available and adhered | A facility was eligible to score a maximum of 3 points, one from each of the following 3 sub-indicators: 1. All pre-determined essential laboratory tests were performed in the past week, 2. All SOPs for the tests available and 3. All randomly selected SOPs are adhered (at least for the past one week). The test menu for laboratory services at PHC is usually long [34]; therefore, we focused on the following important tests during data collection: test for malaria, Hb, urinalysis, UPT, blood glucose level for dispensaries and addition of HIV test to health centres and hospitals. A partial score "0.5" was given if at least 3 tests were performed and "1" given if all mentioned were done. |
| Turnaround time established and monitored | The Facility scored "1" if had established turnaround time for tests performed and it is monitored [34, 35]. |
| Internal Quality control and External Quality Assessment | The facility scored "1" if the laboratory had performed internal quality control and participated in external quality assessment in previous quarter year. Results must have been documented, corrective and preventive actions taken. If either of the two was missing; then "0.5" was given. |
| Laboratory safety system in place | A facility was eligible to score maximum of 3 points, "1" from each of the following 3 sub-indicators: 1. Availability of SOPs for both infection prevention and control (IPC) and post exposure prophylaxis (PEP), 2. Safety rules displayed and 3. Presence of the three colour coded bins with bags to manage healthcare wastes [36, 37]. For health centres and hospitals, an additional requirement was to possess a functional refrigerator for provision of blood bank services. |
| Laboratory supplies management system in place | Facility scored "1" if had documented the incoming and outgoing stock in the register correctly. Three laboratory reagents and or consumable supplies were selected and their records were tracked to determine if were adequately updated. |

We performed univariate linear regression then multiple linear regression to determine the impact of characteristics of PHC facilities on scores for quality of laboratory services. These characteristics are independent categorical variables: Facility's level (hospital/health centre/dispensary), ownership (public or private), location urban or rural) and staffing level (Yes/No).

P-value of < 0.05 was considered significant. Stata version 15 (Stata Corporation, College Station, Texas, USA) was used for data input and statistical calculations.

## Results

### Characteristics of PHC facilities involved in the study

A total of 6,663 PHC facilities were included in this study, whereby 5,485(82.3%) were publicly owned and 1,178(17.7%) were privately owned. The majority of the facilities were the dispensaries 5,721(85.9%) followed by health centres 732 (11.0%) and hospitals level-1 210(3.2%). About one-quarter of the facilities were from urban areas 1,528(22.9%) and the rest were from rural areas 5,135(77.1%).

The average quality score for laboratory services for all indicators across all facilities was 30.8% (SD = 35.7). Approximately one-quarter of the PHC facilities, i.e., 1,773(26.6%) had complied with the laboratory staffing level (Table 2). The compliance was significantly highest at hospitals 40%, followed by health centres 31% and then dispensaries 26% (p<0.001) (Table 2).

The compliance to staffing level was significantly higher in private facilities compared to public facilities 63% vs 19% (p<0.001). Likewise, there was a large compliance gap between urban-based facilities (62%) and rural-based facilities (16%) (p<0.001) (Table 2).

### The proportion of PHC facilities that comply with indicators that measure the quality of laboratory services

As presented in Table 3, the indicator that performed highest (45.1%) in all types of PHC facilities was "facilities with a dedicated department/room for laboratory services" while most of the facilities performed poorly in the implementation of Internal Quality control and External Quality Assessment (22.2%). Less than half of the facilities at the hospital level and health centre level had refrigerators for blood transfusion services. Generally, in all indicators, the compliance improved with the increase in facility level.

### The facility's characteristics that define the scores for laboratory quality services

The findings from multiple linear regression analysis suggest that the characteristics of PHC facilities had a significant effect in predicting scores in the quality of laboratory services among

**Table 2. PHC facilities with appropriate staff (laboratory personnel) for provision of laboratory services on the day of assessment (N = 6,663).**

| Variable | Yes | | No | | Value (chi sq test) |
|---|---|---|---|---|---|
| **PHC facility level** | n | % | n | % | **<0.001** |
| Dispensaries | 1464 | 26 | 4257 | 74 | |
| Health Centres | 225 | 31 | 507 | 69 | |
| Hospitals | 84 | 40 | 126 | 60 | |
| **PHC facility ownership** | | | | | **<0.001** |
| Public | 1,036 | 19 | 4,449 | 81 | |
| Private | 737 | 63 | 441 | 37 | |
| **PHC facility location** | | | | | **<0.001** |
| Rural | 827 | 16 | 4,308 | 84 | |
| Urban | 946 | 62 | 582 | 38 | |
| **All PHC facilities** | 1,773 | 26.6 | 4,890 | 73.4 | |

**Table 3. Proportion of PHC facilities that comply with indicators that measure the quality of laboratory services (N = 6,663).**

| Facilities complied with indicators, n (%) | | Compliance within facility level, n (%) | | |
|---|---|---|---|---|
| Indicator | All facilities (N = 6,663) | Dispensary level (N = 5,721) | Health centre (N = 732) | Hospitals (N = 210) |
| PHC facilities with a dedicated department/room for laboratory services (n = 6,065) | 2,736(45.1%) | 1,912(36.8%) | 628(92.9%) | 196(99.0%) |
| PHC facilities whose laboratory are equipped with necessary equipment | 2,087(34.5%) | 1,360(26.3%) | 535(79.0%) | 192(97.5%) |
| PHC facilities with qualified staff laboratory services (n = 6,663) | 1,773(26.6%) | 1,464(25.6%) | 225(30.7%) | 84(40.0%) |
| Essential laboratory tests provided with SOPs available and adhered | | | | |
| a) PHC facilities with *full-package* of essential laboratory tests (n = 6,118) | 1,976(32.3%) | 1,268(24.2%) | 519(75.7%) | 189(95.5%) |
| b) PHC facilities whose SOPs for all test and equipment use readily available for reference (n = 6,037) | 2,384(39.5%) | 1,640(31.8%) | 564(83.7%) | 180(90.9%) |
| c) PHC facilities whose Laboratory SOPs were adhered (n = 6,039) | 2,272(37.6%) | 1,550(30.0%) | 545(81.1%) | 171(89.4%) |
| PHC facilities with turnaround time established and monitored (n = 6,012) | 1,673(27.8%) | 1,105(21.5%) | 410(61.6%) | 158(80.6%) |
| Internal Quality control and External Quality Assessment conducted (n = 6,052) | | | | |
| a) Both Internal Quality control and External Quality Assessment conducted | 1,342(22.2%) | 774(13.2%) | 416(61.3%) | 152(77.2%) |
| b) Either (only) IQC or EQA is conducted | 839 (13.9%) | 669(12.9%) | 144(21.2%) | 26(13.2%) |
| Laboratory safety system in place | | | | |
| a) Safety/IPC related procedures in place (n = 6,033) | 1,941(32.2%) | 1,274(24.7%) | 491(73.2%) | 176(90.3%) |
| b) Laboratory safety rules displayed (n = 6,032) | 1,953(32.4%) | 1,278(24.8%) | 493(73.4%) | 182(91.9%) |
| c) Healthcare waste management procedures in place (n = 6,027) | 1,751(29.1%) | 1,193(23.1%) | 408(61.1%) | 150(76.9%) |
| d) Hospitals and Health Centres equipped with refrigerator for blood bank services (n = 791) | 363(45.9%) | NA | 194(32.6%) | 167(87.9%) |
| Laboratory supplies management system in place (n = 6,007) | 1,926(32.1%) | 1,301(25.3%) | 460(68.7%) | 165(85.5%) |

Note: A chi-square test was performed on each indicator-facility level pair and p-value of <0.001 was found across all pairs.

PHC facilities. These characteristics explain 57.1% of the variance in laboratory service scores in the model. All characteristic variables showed the same direction of association with the outcome variable for both analysis; univariate (S1 Table) and multivariate (Table 4) linear regression analysis.

The scores for quality of laboratory services were positively defined by compliance to laboratory staffing levels at the facilities (*Beta = 0.430*). On the contrary; the facilities at dispensary level (*Beta = -0.374*), facilities that are located at the rural area (*Beta = -0.070*) and facilities that are owned by public organizations (*Beta = -0.110*) were negatively defined by the scores. The details are shown in Table 4.

## Discussion

The study has shown that the PHC facilities in Tanzania had low level of compliance with all the laboratory indicators. The low performance in laboratory quality indicators has also been reported in other studies. For example, a study in Ethiopia has reported that PHC facilities had inadequate access to essential laboratory tests; [39] and another study also in Ethiopia found that unavailability of laboratory tests in government-owned health centres in southern Ethiopia affected outpatient service utilization causing most diagnoses to be based on clinical findings [40]. A study in four regions in Tanzania found that few dispensaries were providing diagnostic services for HIV.

Generally, performance in laboratory quality indicators in Tanzania is positively linked to the funds provided under vertical disease programs such as HIV and TB [41]. For example; while previous findings from 354 facilities under HIV programmes in Tanzania showed many

**Table 4. The facility's characteristics that define the scores for laboratory quality services at primary healthcare, 2017–2018.**

| Independent variable | B (Slope) | Standard error (SE) | t-ratio(t) | prob.(p) |
|---|---|---|---|---|
| Dispensary level | -0.374 | 0.011 | -34.95 | <0.001 |
| Rural-based facilities | -0.070 | 0.009 | -7.89 | <0.001 |
| Public facilities | -0.110 | 0.009 | -7.89 | <0.001 |
| Complying with lab staffing level | 0.430 | 0.009 | 47.15 | <0.001 |
| Constant, | 0.659 | | | |
| $R^2$ | 0.571 | | | |
| F-ratio F (4, 6658) | 1843.36 | P<0.00001 | | |
| Root MSE | 0.233 | | | |
| N | 6,663 | | | |

$R^2$ = coefficient of determination (adjusted)

facilities were both conducting Internal Quality control (IQA) and participating in External Quality Assessment (EQA) programmes [41]; our findings report undesirable performance in these two indicators in more than third-quarter of the facilities. This present study includes all PHC facilities in Tanzania regardless of funding status.

The critical shortage of laboratory workforce that is recorded in this study is supported by another from a neighbouring country, Rwanda [42] but at the same time contrary to what was found in another neighbouring country; Uganda. In Uganda, there was excess of laboratory workforce in all levels but at hospital level [43]. The healthcare workforce in developing countries is usually concentrated in urban areas and this could be among the reasons why rural-based facilities have been doing poorer compared to urban-based facilities [41]. Shortage of staff is an expensive commodity; addressing it would improve the management of the laboratory quality systems, especially at rural areas where qualified staff are rarely found due to various reasons including staff turnover [44].

Furthermore, both clinicians and clients in developing countries share a common frustration regarding the delay of results from laboratories [42, 45]. This has led to a lack of evidence-based treatment and hence propagates the challenge of Antimicrobial resistance [45]. Improved coordination and supportive supervision could improve the situation [42].

Additionally, we found a critical shortage of laboratory supplies, equipment and inadequate availability of basic tests at laboratory across all facility levels. Verily, the majority of the healthcare facilities from developing countries are struggling to stock adequate equipment and supplies for laboratories due to insufficient budgets [46]. Nevertheless, finances are not a sole contributing factor to chronic stock outs of the items; inadequate laboratory supply chain management plays a major role as well [42]. The challenge of the availability of basic tests goes beyond Tanzania. In Kenya inadequate availability of tests for infectious diseases and other diseases has been shown to affect the readiness of PHC facilities to support the country in implementation of its programme towards the attainment of the UHC target [47]. Odjidja and colleagues have reported that the availability of diagnostics for Malaria, Tuberculosis and HIV is one of the predictors for pregnant women to receive integrated care for all three diseases [48].

## Policy implications

Our study has two important policy implications. First, the results provide a status of country-wide laboratory services quality and capacity in the PHC facilities which will help in the ongoing efforts to strengthen the performance of PHC facilities, as well as in the processes for reimagining PHC services post-COVID-19 [49]. Secondly, based on these findings the

Ministry need to strengthen further the laboratory services in order to be able to support some disease-specific interventions. For instance, in a study conducted in three districts (Uyui, Geita and Ukerewe) in north-western Tanzania involving 80 PHC facilities to assess capacity for integrating schistosomiasis control activities found that only 33.8% (27/80) of the PHC facilities had laboratory services for diagnosis of schistosomiasis [50].

Therefore, the Ministry of Health in collaboration with Ministry responsible for local governments and health sector stakeholders need to further strengthen laboratory services in all PHC facilities, as part of improving and maintaining the quality of laboratory services provided [51]. In doing so, the Ministry will be harnessing its work on the Sustainable Development Goal 3 (*ensure healthy lives and promote well-being for all at all ages*); target 3. 8: "*achieve universal health coverage, including financial risk protection, access to quality essential health-care services and access to . . . . . ..*", [52] since diagnostic services are an essential ingredient of the UHC target [53]. Research from developing countries has shown a strong association between the scope of services provided in PHC facilities and the acceptability of the services offered [13, 54]. Tanzania should consider the establishment of a nationally approved essential diagnostics list (EDL) that will play a role in improving access to essential laboratory tests in PHC facilities.

## Conclusion

The laboratory services quality and capacity in the PHC facilities is still low in Tanzania, characterized by a critical shortage of qualified laboratory personnel especially in public facilities that are based in rural areas. There is a need to further strengthen laboratory services in PHC facilities to ensure quality of laboratory test results generated since even in the laboratories that were found to be working, more than half did not conduct Internal Quality control and or participating in External Quality Assessment. The overall compliance with all laboratory indicators is inadequate (none was achieved by at least half of the facilities). Improvement in laboratory quality indicators such as turnaround time and availability of full-range of basic tests will promote clients' satisfaction; which is among of key measures of quality of healthcare.

## Supporting information

**S1 Table. Univariate analysis of the facility's characteristics that define the scores for laboratory quality services at Primary Healthcare, 2017–2018.**
(DOCX)

## Acknowledgments

The authors thank the Ministry of Health, especially, the Health Quality Assurance Unit for granting us permission to use the SRA data. Aside from government institutions, the authors would like to express their gratitude to development partners the World Bank, the United States Centers for Disease Control and Prevention (CDC), the Danish International Development Agency (DANIDA), and the World Health Organization, who played a crucial role in making SRA possible. Additional acknowledgments go to the communities associated with the visited facilities, PharmAccess International, the Association of Private Health Facilities in Tanzania (APHTA), the Christian Social Services Commission (CSSC), and the Development Partners in Health-Group (DPG-H).

## Author Contributions

**Conceptualization:** Erick Kinyenje, Ruth R. Ngowi, Yohanes S. Msigwa, Joseph C. Hokororo, Chrisogone J. German, Eliudi S. Eliakimu.

**Data curation:** Erick Kinyenje.

**Formal analysis:** Erick Kinyenje.

**Methodology:** Erick Kinyenje, Ruth R. Ngowi.

**Supervision:** Erick Kinyenje, Ruth R. Ngowi, Joseph C. Hokororo, Eliudi S. Eliakimu.

**Validation:** Erick Kinyenje, Ruth R. Ngowi, Joseph C. Hokororo, Chrisogone J. German, Radenta P. Bahegwa, Eliudi S. Eliakimu.

**Visualization:** Erick Kinyenje, Ruth R. Ngowi, Joseph C. Hokororo, Chrisogone J. German, Eliudi S. Eliakimu.

**Writing – original draft:** Erick Kinyenje, Ruth R. Ngowi, Yohanes S. Msigwa, Joseph C. Hokororo, Chrisogone J. German, Radenta P. Bahegwa, Eliudi S. Eliakimu.

**Writing – review & editing:** Erick Kinyenje, Ruth R. Ngowi, Yohanes S. Msigwa, Joseph C. Hokororo, Talhiya A. Yahya, Chrisogone J. German, Akili Mawazo, Mohamed A. Mohamed, Omary A. Nassoro, Mbwana M. Degeh, Radenta P. Bahegwa, Laura E. Marandu, Syabo M. Mwaisengela, Lutengano W. Mwanginde, Robert Makala, Eliudi S. Eliakimu.

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
