## [Decision Letter · Decision Letter 0]

17 Mar 2023

PGPH-D-22-02022

Contribution of quality improvement initiative in strengthening laboratory services in Primary Healthcare Facilities in Tanzania: findings from 2017/2018 Star Rating Assessment

Dear Dr. Kinyenje,

Thank you for submitting your manuscript to PLOS Global Public Health. After careful consideration, we feel that it has merit but does not fully meet PLOS Global Public Health’s publication criteria as it currently stands. Therefore, we invite you to submit a revised version of the manuscript that addresses the points raised during the review process.

Please see the comments below and in the attachment from three reviewers. Please pay special attention to their comments on the statistical analysis. Two reviewers have requested more context on the situation in Tanzania. Please note that although one reviewer suggests shortening the introduction, there are no specific word limits for PLOS Global Public Health.

We look forward to receiving your revised manuscript.

Kind regards,

Hanna Landenmark

Staff Editor

Journal Requirements:

Additional Editor Comments (if provided):

Reviewers' comments:

Reviewer's Responses to Questions

**Comments to the Author**

1. Does this manuscript meet PLOS Global Public Health’s publication criteria? Is the manuscript technically sound, and do the data support the conclusions? The manuscript must describe methodologically and ethically rigorous research with conclusions that are appropriately drawn based on the data presented.

Reviewer #1: Yes

Reviewer #2: Yes

Reviewer #3: No

2. Has the statistical analysis been performed appropriately and rigorously?

Reviewer #1: No

Reviewer #2: No

Reviewer #3: Yes

3. Have the authors made all data underlying the findings in their manuscript fully available (please refer to the Data Availability Statement at the start of the manuscript PDF file)?

Reviewer #1: No

Reviewer #2: No

Reviewer #3: Yes

4. Is the manuscript presented in an intelligible fashion and written in standard English?

Reviewer #1: Yes

Reviewer #2: Yes

Reviewer #3: No

5. Review Comments to the Author

Reviewer #1: The paper presents an important perspective on Quality control of laboratories in PHC settings. However, it needs to address some key issues.

Abstract: Methods should mention the key domains of assessment and score. Results should mention % of dispensary/rural/ and private (given in the main body). Association of total lab score with these factors should be mentioned.

Introduction:

it is too long and needs to be edited to half its length. First two paragraph can be deleted/shortened and full reference to a WHO report is not needed in the main text.

Methods: Current table 1 has to be reorganized. First column can be deleted. And the description of each domain currently in the text can be shifted here as an additional column.

One main problem with the paper is the regression. Since each of the domain score contributes to the total score, they cannot be taken as independent variable. it is no surprise that 99.1% of the variance is explained. It is also apparent from the fact that the domains with more items correlate better to the total score. It is inappropriate to do this. They can keep variables like rural/urban, dispensary/others, private/public etc. in the regression equation.

The total score and domain specific score can be shown (table 4 - first column can be deleted) . Since staff are important inputs one separate table (table 2 & 3 can be combined) can be provided.

Discussion may be rewritten accordingly.

Reviewer #2: 1. Describe the study area. i.e., Tanzania.

1.1. For the global audience to connect with the research, it is important to describe the Tanzania as a nation. This description could be used to put it in the global perspective. For example, if economic indicators like GDP, per capita income are used then it could be used to define if the country is a Low Income country, Middle Income Country or High Income country based on global standards. Some indexes that should be provided are HDI, SDG ranking, Population demographics, social indicators, literacy rate, per capita income, GDP, etc. It could be provided as separate section or sub-section of Methods.

1.2. In the discussion section, compare the laboratory status of Tanzania with other similar nations from literature to better understand the gravity of situation in global context. For example, compare the staffing level status in Tanzania with staffing level in other similar nations. Please provide reasons why it performs better, similar, or worse than other nations.

2. Explain the dependent variable, i.e., SRA.

2.1. In the methods section, please explain the SRA in more detail:

2.1.1. How is it calculated? Do they sum the values from all the 12 criteria? How are the laboratory scores added?

2.1.2. What is the outcome type? Is it a Likert scale, categorical scale or continuous outcome?

2.2. In the results section, please provide the distribution profile of SRA. For example, if it is continuous outcome, please provide mean, median, mode, range, standard deviation, etc. If categorical/Likert, please provide number of datapoints (or, percentage of datapoints) under each category? Is it a uniform distribution for categorical outcome?

3. Please explain why Chi square test is not performed for section “The proportion of PHCs that comply with indicators that measure the quality of laboratory services”?

3.1. Please perform the Chi square test for the section to maintain the consistency in the paper.

4. In Table 4, please provide the overall percentage for following sections:

4.1. 12.2.4

4.2. 12.2.6

4.3. 12.2.7

5. Please provide the facility level distribution of outcomes in Table 4. This could be provided in supplementary section.

6. In the regression model, the r square value of 0.99 needs more clarification.

6.1. Does it indicate that other parameters, i.e., non-laboratory parameters used in preparing the STAR rating do not influence the STAR rating?

6.2. Please perform the univariate analysis with all the independent variables.

6.3. Please prepare the control model with only facility characteristics.

Reviewer #3: The manuscript was presented in such a way that makes it challenging to mention the portion i was talking about. because it would have been nice if all lines were numbered. Some operational definition or explanation in the back ground has to be included to clearly show some one who don't know the system in Tanzania. Other wise the statistical analysis looks good except few comments given in attachment.

6. PLOS authors have the option to publish the peer review history of their article (what does this mean?). If published, this will include your full peer review and any attached files.

**Do you want your identity to be public for this peer review?** For information about this choice, including consent withdrawal, please see our Privacy Policy.

Reviewer #1: **Yes: **Anand Krishnan

Reviewer #2: No

Reviewer #3: No

---

## [Decision Letter · Decision Letter 1]

23 May 2023

PGPH-D-22-02022R1

Status of countrywide laboratory services quality and capacity in Primary Healthcare Facilities in Tanzania: findings from Star Rating Assessment

Dear Dr. Kinyenje,

Thank you for submitting your manuscript to PLOS Global Public Health. After careful consideration, we feel that it has merit but does not fully meet PLOS Global Public Health’s publication criteria as it currently stands. Therefore, we invite you to submit a revised version of the manuscript that addresses the points raised during the review process.

The three previous reviewers were available to comment on your revised manuscript. Two of the reviewers are more positive about this new version, but the other reviewer has raised remaining concerns about your manuscript that need to be addressed. Pay attention to the comment that there have been three previous national surveys on this subject, and contextualise your study in the context of this previous research.

We look forward to receiving your revised manuscript.

Kind regards,

Miquel Vall-llosera Camps

Staff Editor

Journal Requirements:

1. Please send a completed 'Competing Interests' statement, including any COIs declared by your co-authors. If you have no competing interests to declare, please state "The authors have declared that no competing interests exist". Otherwise please declare all competing interests beginning with twhe statement "I have read the journal's policy and the authors of this manuscript have the following competing interests:"

Reviewers' comments:

Reviewer's Responses to Questions

**Comments to the Author**

1. If the authors have adequately addressed your comments raised in a previous round of review and you feel that this manuscript is now acceptable for publication, you may indicate that here to bypass the “Comments to the Author” section, enter your conflict of interest statement in the “Confidential to Editor” section, and submit your "Accept" recommendation.

Reviewer #1: (No Response)

Reviewer #2: All comments have been addressed

Reviewer #3: All comments have been addressed

2. Does this manuscript meet PLOS Global Public Health’s publication criteria? Is the manuscript technically sound, and do the data support the conclusions? The manuscript must describe methodologically and ethically rigorous research with conclusions that are appropriately drawn based on the data presented.

Reviewer #1: Partly

Reviewer #2: Yes

Reviewer #3: No

3. Has the statistical analysis been performed appropriately and rigorously?

Reviewer #1: Yes

Reviewer #2: Yes

Reviewer #3: Yes

4. Have the authors made all data underlying the findings in their manuscript fully available (please refer to the Data Availability Statement at the start of the manuscript PDF file)?

Reviewer #1: Yes

Reviewer #2: No

Reviewer #3: Yes

5. Is the manuscript presented in an intelligible fashion and written in standard English?

Reviewer #1: Yes

Reviewer #2: Yes

Reviewer #3: Yes

6. Review Comments to the Author

Reviewer #1: Introduction is too long and needs to be condensed to half. Unnecessary details in methods section on SDGs/HDI is not required.

The paper uses data from a report, probably in public domain, and the value of addition of this paper is not clear. It is also understood that there have been three such national surveys with the most recent being 2020-21. I think the value addition would be to see the change in the laboratory services in Tanzania over time and link it to the interventions. Without that this paper publication value is low.

In most places, scores are ‘0’ and ‘1’. Could the authors consider them as nothing – 0, partial – 1 and complete – 2 to give better sensitivity to the score.

Table 1 is not required.

The outcome variable for the linear regression was the crude score or the percentage score out of maximum. It should be noted that dispensary had a lesser maximum score than the health centers and hospitals.

Reviewer #2: Almost all the modifications have been done in the article. Thus, it can be accepted. However, I would like to provide mere suggestions based on more clarity provided in the manuscript. No additional review is needed if authors choose to work on these suggestions.

• Author is encouraged to provide the control model, it could be provided as supplementary file, which will help the readers to get the complete picture of the variables

• Authors research is addressing a key issue in healthcare system. Laboratories need the focus in the healthcare system and policies should be more proactive. However, the idea “diagnostic services are an essential ingredient of the UHC target” could be strengthened with some references. One of the key ideas to strengthen this argument is to provide some quantitative relationship between laboratory and policy goals (sub-national, national or international). For example, in the following papers authors have tried to establish a quantitative relationship between laboratories and overall facility/policy performance.

o https://www.emerald.com/insight/content/doi/10.1108/09564230010340715/full/html

o https://www.ncbi.nlm.nih.gov/pmc/articles/PMC6635801/

Authors are encouraged to refer these or similar research papers in their policy recommendation section to provide a stronger justification for the need to focus on the issues identified by the authors.

Reviewer #3: The authors have gone through each comments well and revised what they need to. Now what may be need to added is just they need to re arrange it to the publication format which of course the editors will go through. other wise i am comfortable with the paper.

7. PLOS authors have the option to publish the peer review history of their article (what does this mean?). If published, this will include your full peer review and any attached files.

**Do you want your identity to be public for this peer review?** For information about this choice, including consent withdrawal, please see our Privacy Policy.

Reviewer #1: **Yes: **Anand Krishnan

Reviewer #2: No

Reviewer #3: **Yes: **Abela, Abebe Negesso

---

## [Decision Letter · Decision Letter 2]

15 Aug 2023

PGPH-D-22-02022R2

Status of countrywide laboratory services quality and capacity in Primary Healthcare Facilities in Tanzania: findings from Star Rating Assessment

Dear Dr. Kinyenje,

Thank you for submitting your manuscript to PLOS Global Public Health. After careful consideration, we feel that it has merit but does not fully meet PLOS Global Public Health’s publication criteria as it currently stands. Therefore, we invite you to submit a revised version of the manuscript that addresses the points raised during the review process.

The minor comments by our third reviewer, mentioned at the bottom of this email, were left answered in the last version, therefore you are particularly encouraged to address those.

We look forward to receiving your revised manuscript.

Kind regards,

Shifa S. Habib

Academic Editor

Journal Requirements:

Additional Editor Comments (if provided):

Reviewers' comments:

Reviewer's Responses to Questions

**Comments to the Author**

1. If the authors have adequately addressed your comments raised in a previous round of review and you feel that this manuscript is now acceptable for publication, you may indicate that here to bypass the “Comments to the Author” section, enter your conflict of interest statement in the “Confidential to Editor” section, and submit your "Accept" recommendation.

Reviewer #1: All comments have been addressed

Reviewer #2: All comments have been addressed

Reviewer #3: (No Response)

2. Does this manuscript meet PLOS Global Public Health’s publication criteria? Is the manuscript technically sound, and do the data support the conclusions? The manuscript must describe methodologically and ethically rigorous research with conclusions that are appropriately drawn based on the data presented.

Reviewer #1: Yes

Reviewer #2: Yes

Reviewer #3: No

3. Has the statistical analysis been performed appropriately and rigorously?

Reviewer #1: Yes

Reviewer #2: Yes

Reviewer #3: Yes

4. Have the authors made all data underlying the findings in their manuscript fully available (please refer to the Data Availability Statement at the start of the manuscript PDF file)?

Reviewer #1: Yes

Reviewer #2: Yes

Reviewer #3: Yes

5. Is the manuscript presented in an intelligible fashion and written in standard English?

Reviewer #1: Yes

Reviewer #2: Yes

Reviewer #3: No

6. Review Comments to the Author

Reviewer #1: None

Reviewer #2: All my queries have been addressed. No more changes suggested.

Reviewer #3: Most of my previous version comments were not addressed well or given justification for not changing it. The Authors are expected to address the comments before third submission for me just not to waste time commenting same thing always.

7. PLOS authors have the option to publish the peer review history of their article (what does this mean?). If published, this will include your full peer review and any attached files.

**Do you want your identity to be public for this peer review?** For information about this choice, including consent withdrawal, please see our Privacy Policy.

Reviewer #1: **Yes: **Anand Krishnan

Reviewer #2: No

Reviewer #3: No

Reviewer 3 comments:

The manuscript was difficult to comment as the lines were not numbered just as the the older version.Again as just last version comment which is not corrected by the authors is study population should tell us. which health facilities are included, from which health facilities data extraction taken place? How the selection of PHC were taken place, what method was used to identify this health facilities from whole? if the authors took the whole health facilities in the country it okay and mention it otherwise still the explanation it needs correction.Does qualified laboratory have operational definition? What is the criteria to say qualified? Operationalize it on this paper please.to label qualified which or how many indicators are expected to fulfil?

---

## [Editor Report · Decision Letter 3]

26 Sep 2023

Status of countrywide laboratory services quality and capacity in Primary Healthcare Facilities in Tanzania: findings from Star Rating Assessment

PGPH-D-22-02022R3

Dear Dr. Kinyenje,

We are pleased to inform you that your manuscript 'Status of countrywide laboratory services quality and capacity in Primary Healthcare Facilities in Tanzania: findings from Star Rating Assessment' has been provisionally accepted for publication in PLOS Global Public Health.

Best regards,

Shifa S. Habib

Academic Editor
